# Are Sex Differences in Collegiate and High School Sports-Related Concussion Reflected in the Guidelines? A Scoping Review

**DOI:** 10.3390/brainsci13091310

**Published:** 2023-09-12

**Authors:** Patryk A. Musko, Andreas K. Demetriades

**Affiliations:** 1College of Medicine and Veterinary Medicine, University of Edinburgh, Edinburgh EH16 4SB, UK; patrykmusko@icloud.com; 2Department of Neurosurgery, Royal Infirmary Edinburgh, Edinburgh EH16 4SA, UK

**Keywords:** sports-related concussion, sex differences, concussion guidelines, concussion in sports group, concussion, traumatic brain injury, chronic traumatic encephalopathy, sports injury

## Abstract

**Background**: Sport-related concussion (SRC) is a common sport injury. Females are participating in sports at increasing rates, and there is growing awareness that female athletes may be more vulnerable to SRC. **Objectives**: We aimed to identify sex differences in epidemiology, clinical manifestation and assessment of SRC and examine how these relate to the 6th International Conference on Concussion in Sport (ICCS). **Methods:** We conducted a scoping review of the Medline database and identified 58 studies examining the effects of sex on SRC in collegiate and high school athletes that were written in English and published in a peer-reviewed journal between March 2012 and March 2022. **Results**: We found that female athletes suffer higher rates of concussion in sex-comparable sports, in particular soccer. Female athletes experience more somatic symptoms—headache/migraine/sleep disturbance—and may take longer to recover from concussion. Sex differences were also identified regarding some aspects of sideline concussion assessment with the Sport Concussion Assessment Tool. **Conclusions:** Females are at greater risk and experience SRC differently than males; this is mostly likely due to a combination of biomechanical factors, differences in neck musculature and hormonal and social factors. Sex differences are not widely addressed by the 6th ICSS, which informs many sports’ concussion protocols.

## 1. Introduction

Sport-related concussion has been described as a ‘traumatic brain injury that is defined as a complex pathological process affecting the brain, induced by biomechanical forces with several common features that help define its nature’ by the Concussion in Sports Group (CISG) [1]. In recent years, SRC has been the focus of increased attention from both medical professionals and lay people due to increased recognition of the associated sequelae, most notably chronic traumatic encephalopathy and neurodegenerative disease [2]. During this time, female participation in both amateur and professional sports has also been increasing [3], with females now making up 43.7% of athletes within all divisions of the National Collegiate Athletic Association (NCAA) [4]. Alongside the increase in female participation in sports, there is a growing awareness that anatomical, physiological and biomechanical factors place female athletes at greater risk of sustaining sports injuries, including SRC [4]. Despite this, female athletes are underrepresented in the SRC literature, including in consensus statements on SRC [5].

Numerous organizations have worked on developing consensus and position statements to provide evidence-based recommendations for the prevention and management of SRC. Arguably one of the most influential groups to publish SRC consensus statements is the CISG, which held its most recent 6th International Conference in Sport in Amsterdam in 2022 [6]. During the conference, 13 ‘Rs’ aimed at guiding the management of SRC were developed: Recognize, Reduce, Remove, Refer, Re-evaluate, Rest, Rehabilitate, Recover, Return to learn/Return to sport, Reconsider, Residual Effects, Retire and Refine [6]. Recommendations set out in the fifth iteration of the CISG’s consensus statement have been implemented in the concussion protocols of numerous professionals and amateur sport governing bodies including the National Football League, World Rugby and English Football Association [7,8,9]; however, there is yet to be any study published on the implementation of the sixth consensus statement.

The aim of this study was to perform a scoping review on the topic of sex differences in SRC to investigate and map the current literature on the topic. Our primary aim was to identify the sex differences in the epidemiology of SRC, underlying reasons why females may be more vulnerable to SRC, differences in symptoms experienced and time to recover from concussion between male and female athletes, and finally sex differences in concussion assessment. Our secondary aim was to examine whether any sex differences we identified were addressed by the CISG’s sixth consensus statement. We felt this was warranted as few studies have evaluated concussion guidelines in relation to sex differences [10]. We opted to focus on investigating sex differences in high school (HS) and collegiate athletes as this is the demographic that is most affected by SRC [11].

Within the literature on SRC, gender and sex are often used interchangeably; however, it is important to highlight that the term ‘sex’ is used to refer to the biological differences between males and females, whereas ‘gender’ refers to socially constructed roles, behavior and identities [12]. Sex and gender are both known to play a role in concussion [13]; however, in this review, we were primarily interested in investigating biological differences in SRC, and therefore the term ‘sex’ is used throughout.

## 2. Materials and Methods

The scoping review used the methodology as per Arksey and O’Malley’s five-stage scoping review process [14] and Preferred Reporting Items for Systematic Reviews and Meta-Analyses (PRISMA)-Scoping Review checklist [15].

### 2.1. Identification of the Research Question(s)

The following research questions were identified for this review:Are there sex differences in the epidemiology of SRC?Are there sex differences which cause females to be more vulnerable to SRC?Are there sex differences in sideline SRC assessment or neurocognitive testing (NCT)?Are there sex differences in symptoms of SRC and/or recovery time?Do sex differences impact the recommendations set out in the Amsterdam Consensus on SRC?

### 2.2. Identification of the Relevant Studies

A literature search using the PubMed search engine to identify relevant studies in the Medline database was conducted using keywords and phrases identified during an initial search of the literature. The search strings used to identify relevant studies were ‘Sex’ AND ‘Concussion’; ‘Sex’ AND ‘Sport’ AND ‘Concussion’; ‘Gender’ AND ‘Sport’ AND ‘Concussion’; and ‘Gender’ AND ‘Concussion’.

### 2.3. Study Selection

The titles and abstracts of the identified studies were screened and included in this study if the following inclusion criteria were met: (1) written in English and published in a peer-reviewed journal between March 2012 and March 2022 and (2) compared differences between male and female high school (HS) or collegiate athletes. Exclusion criteria were (1) only a single sport studied; (2) concussions acquired outside of sports; (3) focused on severe traumatic brain injuries; (4) imaging or biomarker studies; (5) case reports and narrative reviews.

### 2.4. Charting the Data

Data from the relevant studies were collated using Microsoft Excel (version 2208) with the following data extracted from selected studies: author, study design and key sex differences identified.

### 2.5. Collating, Summarizing and Reporting the Results

Results from this study were presented as a PRISMA diagram (Figure 1), and a qualitative descriptive approach was used to summarize key findings as they related to sex differences in epidemiology, risk factors, symptoms, time to recovery (TTR) and NCT.

## 3. Results

In total, 58 studies that meet the inclusion criteria were identified during the literature search.

**Figure 1 brainsci-13-01310-f001:**
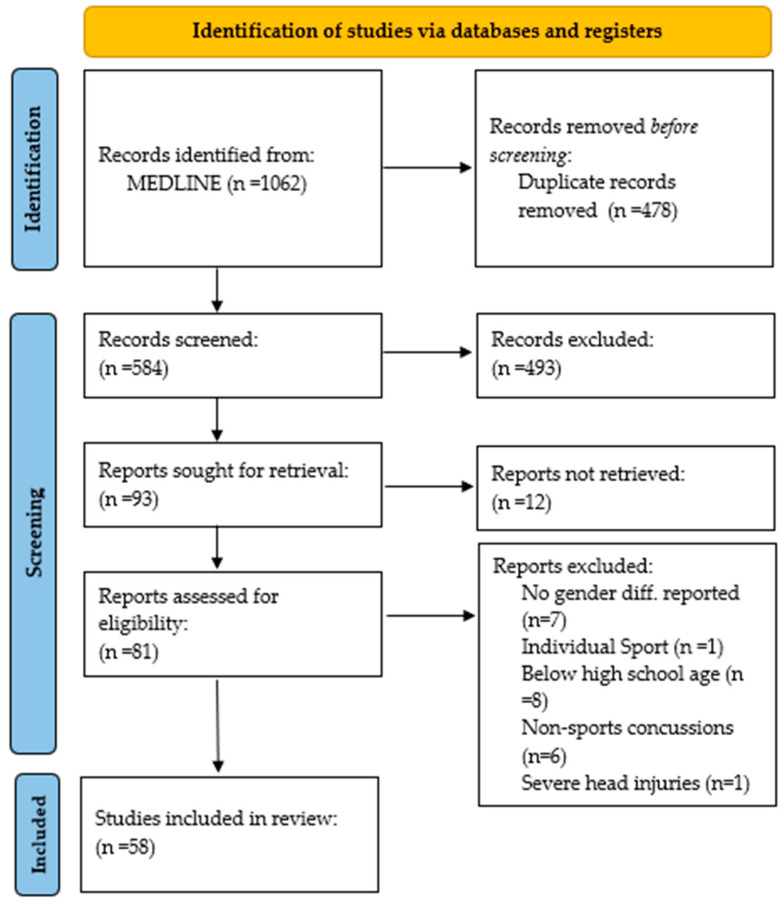
PRISMA 2020 flow diagram for systematic reviews (own work). Template: Page et al. [16].

### 3.1. Sex Differences in Epidemiology of Sport-Related Concussion

A total of 12 studies examined sex differences in the epidemiology of SRC. A summary of each study is shown in Table 1.

Large-scale descriptive epidemiological studies using data from the National Collegiate Athletic Association’s Injury Surveillance Program revealed that the incidence of SRCs was highest in men’s ice hockey, followed by women’s soccer. In sex-comparable sports (soccer, basketball, softball/basketball and swimming/diving), females had higher rates of concussion [17]. The Ivy League–Big Ten Epidemiology of Concussion Study found that out of 27 sports, women’s lacrosse had the highest incidence of concussion, followed closely by men’s football. Additionally, females sustained higher rates of concussion in soccer, ice hockey and basketball [18]. A study on Columbia University athletes found that 23% of female athletes experienced SRC compared to 17% of male athletes [19]. Data from the National HS Reporting Information Online Injury Surveillance System show that men experience a higher total number of SRCs; however, in sex-comparable sports, women sustain SRCs at higher rates, with lacrosse being a notable exception [20]. Consistently, studies found that concussion rates were highest in American football, followed by women’s soccer [20,21,22,23]. Studies using data from HSs in Michigan and Hawaii found similar results, with female athletes experiencing a higher incidence of SRC in all sex-comparable sports [24], apart from lacrosse [25,26].

A meta-analysis of SRC epidemiology in HS and collegiate athletes by van Pelt et al. [27] found that women sustained only 0.11 more concussions per 10,000 athletic exposures (Aes); however, this was statistically significant. The study also concluded that females had a statistically significantly higher SRC incidence in soccer, basketball, volleyball and baseball/softball. Lacrosse was the only sport where the incidence was significantly higher in males. A meta-analysis by Cheng et al. [28] found similar results; however, only women’s soccer and basketball had a significantly higher concussion rate. Again, it is worth noting that, while not statistically significant, men’s lacrosse was found to have a higher concussion rate. The discrepancies in significance between the two studies might be explained by differences in the demographics of the articles analyzed.

### 3.2. Sex-Based Vulnerabilities for Sport-Related Concussion

We identified a total of 18 studies examining sex differences that might result in women being more vulnerable to SRC. A summary of each study is shown in Table 2.

A systematic review by McGroaty et al. [29] identified that differences in biomechanics and neck strength, hormonal influences and psychosocial factors may all have a role in explaining the sex differences seen in SRC. A study investigating differences in neck size in incoming collegiate athletes found that females had significantly smaller necks, both overall and in proportion to body mass index. However, there was no association between neck size and previous SRC [30]. A study using head impact sensors during HS sports found that female athletes experienced significantly fewer head impacts per 1000 AEs compared to males when playing soccer, basketball and lacrosse [31]. One study found females were less likely to suffer from concussions due to contact with other players in lacrosse, basketball, ice hockey and soccer; however, they were more likely to sustain concussion while heading the ball in soccer or after coming into contact with the ball or equipment in lacrosse and soccer [32]. A single study investigating hormonal influences was identified, which found that among concussed female collegiate athletes, menstrual cycle phase at time of injury was not associated with differences in symptom severity or time to recovery (TTR). However, hormonal contraception use was found to result in significantly less severe post-concussion symptoms compared to non-hormonal contraception users, although there was no effect on TTR [33]. Studies have identified that females are more anxious about SRCs [34,35] and may have greater knowledge regarding concussion [36]. Compared to males, females display more traits that have been linked to a higher likelihood of concussion disclosure [37,38,39,40]. Regardless, few studies have proven that females are more likely to disclose SRC symptoms [36,41], with one study showing that male sex is associated with non-disclosure while female sex is not a significant factor in non-disclosure of concussive symptoms [42]. Additionally, some studies have not found any sex differences in concussion attitudes, beliefs, reporting intentions [43,44] or behaviors [45].

Finally, one study found that females may be at higher risk of not being removed from play immediately following concussion [46].

### 3.3. Sex Differences in Concussion Assessment

A total of 17 studies examined sex differences in SRC assessment and NCT both at baseline and post-injury. A summary of each study is shown in Table 3.

A study investigating concussion assessment using the Sport Concussion Assessment Tool 5 (SCAT5) found no sex differences for any SCAT5 component at baseline testing [47]. Norheim et al. [48] and Hutchison et al. [49] both found that at baseline, females performed better than males on both the immediate and delayed 10-word-list portion of SCAT5. Bailey et al. [50] observed that when SCAT5 was administered at baseline, females reported slightly more severe concussion-like symptoms. Additionally, when a threshold SCAT5 standardized assessment (SAC) was used to predict SRC, accuracy was improved with the use of sex-normative data. It is worth noting that SAC change from baseline was found to be more accurate in diagnosing SRC and was not affected by the use of sex-normative data [50]. Sex differences in previous iterations of SCAT5 (SCAT3) were also investigated, with one study concluding that post-concussion females reported higher symptom severity scores immediately after being concussed, but total SAC scores did not vary between the sexes [51]. Another study identified that at baseline, females performed better on a cognitive assessment, attaining higher scores on SAC, delayed recall and orientation. Post-concussion orientation was the sole component in which females performed significantly better. While statistically significant, these sex differences were felt to be too small to be clinically significant. Clinically significant differences were observed for self-reported symptoms, with fewer females reporting neck pain at baseline and experiencing less severe visual blurring and irritability post-injury [52].

We identified three studies investigating sex differences in NCT with Immediate Post-Concussion Assessment and Cognitive Testing (ImPACT). One study concluded that there were no sex differences in total scores at baseline or following SRC in HS athletes; however, when a summary score consisting of multiple NCT batteries was used, females were significantly more cognitively impaired following SRC [53]. A study on collegiate athletes found that at baseline, females performed better on the visual motor speed component of ImPACT; however, there were otherwise no significant differences in scores at baseline or post-SRC nor differences between baseline and post-concussion scores [54]. A single study found that female athletes reported more symptoms and scored worse on measures of visual memory following SRC [55]. No sex differences were observed in NCT using the King–Devick assessment [56]. Females at baseline and with a history of SRC performed worse on executive function and memory when using the Cogstate battery; however, there was no compounding effect of concussion. Females who had suffered a previous SRC performed worse on the Cogstate assessment with a two-back condition which increases cognitive load but is not included within the standard test [57]. Studies also examined sex differences in the assessment of concussion using gait, with no sex differences observed in tandem-gait tasks [58] or single-task gait recovery; however, females had longer dual-task gait recovery [59]. Sex differences observed in vestibular ocular motor screening [60,61] and autonomic regulation [62,63] do not seem to be consistent among studies.

### 3.4. Sex Differences in Symptoms and Recovery Time

Sex differences in symptoms experienced and TTR were described in 11 studies. A summary of each study is shown in Table 4.

Female HS athletes have been shown to experience more severe symptoms acutely post-concussion [64] and have higher scores for anxiety/mood, cognition/fatigue and ocular symptoms [65]. Additionally, females involved in HS high-contact sports were more likely to report irritability and less likely to experience loss of consciousness. In low-contact sports, females were less likely to experience amnesia following SRC [66]. The literature on reported TTR is mixed; however, the same study identified that fewer females than males in high-contact sports experienced symptom resolution in fewer than 7 days; in addition, a greater percentage of females took longer than 21 days to recover from symptoms or return to play [66]. This is supported by Bretzin and colleagues [67], who found that female collegiate athletes in both low- and high-contact sports took longer on average to return to play and academics. Other studies found no significant sex differences in TTR in collegiate athletes [68,69] or development of post-concussion syndrome [70]. The literature on sex differences in return to sport exercise programs found that concussed HS females took longer on average to complete a six-step graded exercise program [71]. Another study found that concussed HS athletes prescribed an aerobic exercise program recovered faster than those prescribed either rest or a placebo-like stretching program, and there were no significant differences in TTR between males and females in any of the groups [72]. A systematic review by Iverson et al. [73] on characteristics associated with prolonged recovery concluded that while the literature is inconsistent, it suggests females experience a higher number and severity of symptoms and take longer to recover.

One study found that female athletes were more likely to be prescribed medications following SRC [74].

## 4. Discussion

### 4.1. Sex Differences in Sport-Related Concussions

This scoping review has identified that there are sex differences in the epidemiology, concussion assessment, symptoms experienced and TTR of SRC. Female HS and collegiate athletes are at increased risk of sustaining SRC, especially in soccer and basketball. While males were more likely to experience SRC in lacrosse, this might be explained by differences in the rules of the game, since body checking is only permitted in men’s lacrosse [75]. This study has identified the reasons why females may be at increased risk of concussion. Multiple studies have identified sex differences in neck musculature as a risk factor for SRC [76,77]; we found that while females do have smaller neck sizes, this was not associated with a history of concussion. Despite this, it is interesting to note that studies focusing on the role of cervical muscles during heading in soccer have shown that players with a smaller neck size and lower strength experience greater head acceleration [78]; this may in part explain why women’s soccer was frequently found to have the highest rate of concussion out of all sex-comparable sports. There is, however, evidence that neck muscle size and strength is not the only factor involved in responding to head impact, with some evidence showing that there is a difference between how male and female cervical muscles react to head impacts. Females have been shown to activate their sternocleidomastoid muscles earlier in response to impact, resulting in greater peak angular acceleration and displacement [79]. Females having a lower biomechanical tolerance to head impacts [80] may explain why we observed that despite experiencing fewer head impacts than males per AE, females were consistently found to have higher rates of concussion in soccer and basketball.

Females may also be more vulnerable to SRC due to hormonal influences. Researchers have proposed a ’withdrawal hypothesis’, according to which head injury during the high progesterone luteal phase of the menstrual cycle leads to disruption of hypothalamic and pituitary function, precipitating a rapid fall in progesterone levels [81,82]. However, evidence for this primarily stems from animal models in which progesterone and estrogen have been shown to be neuroprotective; meanwhile, evidence supporting this hypothesis in humans is limited [83]. Despite contradictory evidence, studies agree female athletes do seem to be more likely to disclose SRC than males, posing a challenge to studying sex differences in SRC as this is more likely influenced by gender identity and social factors, rather than biological differences [84].

The literature on sex differences in SCAT5 testing at baseline and post-injury is varied; however, it seems to suggest that even though sex may impact specific components, such as SAC or 10-word list, there are no differences in overall SCAT5 scores. While there are limited studies on sex differences in SCAT5, studies using previous iterations (SCAT3/2) seem to support our findings [85].

Studies have suggested that there is a sex difference in the constellation of symptoms following SRC, with fatigue/sleep disturbance, difficulty concentrating and emotional instability reported more frequently by females [82,86,87,88]. This is widely in line with our findings. It is worth noting that these symptoms are all observed in premenstrual syndrome, potentially further suggesting a hormonal influence [89]. While not consistent among all studies, our findings do seem to suggest that female athletes seem to take longer to recover from concussion; however, this is not consistent among all studies. This is supported by the American Medical Society for Sport Medicine position statement, which lists female sex as a risk factor for prolonged concussion [90]. One potential explanation for prolonged TTR is that females report a higher number and greater severity of symptoms due to the factors mentioned above, such as a lower biomechanical tolerance to impacts and hormonal factors, and thus there is a larger deficit to recover from [71].

### 4.2. Are Sex Differences Reflected in the Amsterdam Consensus

The CISG consensus statement on concussion in sports does not make any recommendations for the management of concussed female athletes, with younger athletes being the sole demographic for which specific recommendations are made [6]. However, the sex differences examined in this study may be relevant to the 13 ‘Rs’ set out by the CISG.

Within ‘Recognize’, the *CISG* outlines their proposed definition of SRC [6], as stated within the Introduction of this paper. We did not find any evidence in our study that would support the need for the definition of SRC to vary by sex. As part of ‘Reduce’, the *CISG* proposes strategies to reduce concussions. These include specific rule changes in child and adolescent ice hockey, and the success of banning body checking in youth ice hockey is emphasized. Within the consensus statement, no policy recommendations are made for female sports; however, we found that female athletes are at high risk of concussion in soccer due to heading the ball, and therefore it may be worthwhile to consider policy changes in order to mitigate these risks. Similar policies prohibiting heading in youth soccer are already being considered [91]. Other concussion prevention strategies include the use of protective equipment and neuromuscular training warm-up programs; however, we did not find any studies examining sex differences in either of these concussion prevention strategies. Within the consensus statement, however, it was felt that more research is required on the use of neuromuscular warm-up training programs in women’s sports [6]. As previously discussed, sex differences in cervical muscles may place females at a biomechanical disadvantage in regard to head impacts; therefore, it would be reasonable to suggest that this risk factor could be modified through neck-strengthening programs. However, in the consensus, further research is encouraged on this topic, and there are no recommendations made for neck-strengthening programs for male or female athletes [92]

Within ‘Remove’, it is recommended that all athletes with suspected SRC be immediately removed from play [6] and assessed at the sidelines, with SCAT5 thus far being the preferred test [93]. Interestingly, we found one study which shows that concussed female athletes are less likely to be removed from play; however, more research is needed to determine underlying causes for this. While current studies suggest SCAT5 overall scores are not affected by sex, the use of sex-normative baseline scores may improve diagnostic accuracy. Overall, there are limited data on sex differences in SCAT5. Within the sixth consensus statement, the *CISG* proposed modifications to SCAT5 in order to develop an updated assessment tool (SCAT6). These changes included removing vestibular–ocular motor screening, adding dual-gait tasks and expanding the 10-word list, among other changes [6]. The *CISG* has stated that expanding the 10-word list to a 12- or 15-word list will improve psychometric properties [6], as female athletes appear to perform better on the 10-word list. It will be interesting to observe whether longer word lists will yield more comprehensive results when assessing SRC in females. Demographic differences were also observed by the *CISG* when creating SCAT6 [94], and further work is required to fully investigate sex differences in concussion assessment with SCAT6.

In ‘Re-evaluate’ the *CISG* announces a new Sport Concussion Office Assessment Tool (SCOAT6) to be used from 72 h to several weeks post-concussion to guide more individualized management plans [4]. Similar to SCAT6, there is yet to be any study on the effect of sex on SCOAT6. The *CISG* also makes recommendations regarding the use of NCT and states that while it may be useful for certain demographics, such as professional and elite athletes, they discourage the routine use of NCT [89]. Therefore, while some studies suggest there may be sex differences in some NCT batteries, these may not be too meaningful for clinical practice.

In the 2022 consensus, the *CISG* once again found that the previous recommendation of strict rest following SRC is not beneficial and instead supports a gradual return to physical activity [6]. These recommendations are outlined as part of ‘Rest’. In our paper, we included one review that did not find any sex differences in athletes prescribed exercise post-concussion; however, there is an overall lack of studies examining the role of sex in exercise following concussion [95].

Within ‘Refer’, the *CISG* defines ‘persisting symptoms’ as those lasting for more than 4 weeks in all age groups and suggests those suffering from persisting symptoms be referred for a multimodal assessment, preferably by a multidisciplinary team [6]. In the previous 2016 Berlin Consensus [96], persistent symptoms were defined as those occurring beyond 14 days following injury; however, females seem to take longer to recover, with some studies suggesting symptoms may persist for up to a month [73]. This amended recovery time, therefore, is more reflective of the clinical picture of SRC in female athletes.

Within ‘Rehabilitation’, the CISG recommends cervicovestibular rehabilitation for neck pain and/or headaches persisting for longer than 10 days and vestibular rehabilitation for symptoms such as dizziness/balance problems [6]. In our study, we did not find any consistent sex differences in neck pain or vestibular symptoms, which suggest females would benefit more or less from cervicovestibular or vestibular rehabilitation. Within ‘Recover’, the *CISG* outlines suggestions for the clinical assessment of recovery, which involves the assessment of symptoms, time taken to return to learn and return to play and other outcome measures, such as response to physical activity, post-traumatic headaches, assessment of balance and VOMS screening [6]. As previously mentioned, while not consistent across all studies, we found that females appear to have a longer TTR. However, in this study, we mainly used resolution of symptoms and time taken to return to play/return to learn as measures of recovery; therefore, future work should also aim to examine whether there are sex differences in recovery using outcomes such as those listed above. Within ‘Return-to-play’/’Return-to-learn’, the *CISG* agrees that females have a longer TTR, but state that most cohorts including female athletes have similar recovery patterns and can be managed with similar recovery strategies [97].

Within ‘Reconsider’, the long-term sequalae of SRC are discussed. While these were beyond the scope of this paper, there is some evidence that sex may have a role in the development of long-term consequences [98]. In ‘Retire’, the *CISG* states that no factors have been identified that, if present, would require an athlete’s retirement from sport following SRC [6]. In keeping with this, we did not find any studies on sex differences in recruitment from sports following SRC.

Within ‘Refine’, considerations to strengthen the consensus process are outlined and special considerations are outlined for para and pediatric athletes; however, there are no separate considerations made for female athletes [6].

### 4.3. Limitations and Future Work

This study had multiple limitations which should be considered when interpreting the results. Firstly, due to lack of access, 12 full-text articles identified during the literature search could not be retrieved. Another limitation of our study was that only the Medline database was searched for relevant articles, and hence there is a possibility that studies indexed in other databases were omitted, potentially affecting the data and subsequent conclusions. Finally, we only included studies published between March 2012 and March 2022; therefore, we did not examine the full scope of the literature on sex differences in SRC.

Here, we focused on sex differences in HS and collegiate athletes; however, we did not examine differences in SRC between these two groups or the confounding effect of age [6,89,99]. Furthermore, our study did not consider sex differences in other demographics, such as children or elite athletes.

As previously mentioned, differences in individual sports’ rules may impact SRC. In addition, position played [100] may have an impact on SRC. There are limited data on sex differences in individual sports, and so this was not considered here. Due to this, our finding cannot be used to guide concussion guidelines for individual sports. Another limitation is the bias of reporting found in SRC research, with an overwhelming number of studies focusing on athletes in HS and colleges within the United States. However, as concussions occur worldwide, it would be difficult to comment on sex differences in concussions in countries where there may be differences in sports played, management principles and sociodemographic differences. This limitation was likely further exacerbated due to our methodology, as we only included studies written in English. Finally, although we were examining sex differences, only one study [50] included reported participants’ gender identity and biological sex.

Future work on the topic of SRC should focus on further researching the effect of sex on symptom severity and duration of symptoms and aim to disentangle whether observed differences are biological or influenced by social attitudes on symptom reporting. To do so, future studies on SRC should follow the sex and gender equity research guidelines [12] and report both sex and gender separately when feasible. We would also implore future studies to look into investigating in greater detail the sex differences in cervical muscles on SRC and whether strategies such as neck training programs or the use of neuromuscular warm-up programs are useful in preventing SRC. Finally, we would also recommend more research on the sex differences in exercise programs following SRC in order to allow clinicians to understand how best to tailor SRC recovery programs for concussed females.

## 5. Conclusions

Overall, we have shown that female athletes experience a higher incidence of concussion in sex-comparable sports and suffer from more severe and longer-lasting symptoms. There is some evidence that sex may also play a role in performance on individual aspects of concussion testing with SCAT; however, overall scores do not seem to be impacted by sex. Sex differences in SRC are likely multi-factorial, but there is evidence that biomechanics and neck strength, hormonal influences and social factors may all be relevant. However, at present, the international consensus on concussion in sports, which informs many sports’ concussion protocols, does not comprehensively account for sex differences.

Future research is encouraged in order to unravel the complex biomechanical, hormonal and social influences which are implicated in sex differences in SRC. A greater understanding of these will hopefully allow for evidence-based prevention and management protocols for female athletes. We also encourage future work to focus on improving female representation within SRC research.

## Figures and Tables

**Table 1 brainsci-13-01310-t001:** Summary of studies examining sex differences in SRC epidemiology.

Author(s), Year	Study Design	Key Findings
Chandran et al., 2021 [17].	Descriptive Epidemiological	In sex-comparable sports, females sustained higher concussion rates (basketball, soccer, softball and swimming/diving). Females were more likely to sustain concussions due to contact with equipment/apparatus.
Putukian et al., 2019 [18].	Descriptive Epidemiological	Women’s lacrosse has the highest rate of SRC (1.35 per 1000 AEs ^1^), followed by American football (1.26 per 1000 Aes). Females experienced higher SRC rates in soccer (1.07 vs. 0.94 per 1000 Aes), ice hockey (0.96 vs. 0.68 per 1000 Aes) and basketball (0.57 vs. 0.43 per 1000 Aes)
Davis-Hayes et al., 2017 [19].	Retrospective Cohort	A total of 23% of female athletes experienced at least one SRC compared to 17% of males (*p* = 0.001). Females were more likely to sustain SRCs in soccer.
Haarbauer-Krupa et al., 2018 [20].	Descriptive Epidemiological	The SRC rate was highest in American football (9.0 per 1000 Aes), followed by women’s soccer (5.8 per 1000 Aes). Females had higher rates of SRC in all sex-comparable sports.
Kerr et al., 2019 [21].	Descriptive Epidemiological	American football had the highest SRC rate (10.40 per 10,000 Aes), followed by female soccer (8.19 per 10,000). Females experienced 3.35 concussions per 10,000 Aes compared to males, who experienced 1.51 per 10,000 Aes.
Schallmo et al., 2017 [22].	Descriptive Epidemiological	Females had a higher incidence of SRC (*p* < 0.05). Men’s football (*p* < 0.0001) and women’s soccer (*p* = 0.0002) had the highest SRC rates.
Yang et al., 2017 [23].	Interrupted time series	Male athletes sustained an overall greater number of SRCs (*p* < 0.001). Females sustained almost double the SRCs in sex-comparable sports (36.1 vs. 18.1 per 1000 Aes, *p* < 0.001). American football had the highest annual concussion rate (7.84 per 10,000 AE), followed by female soccer (5.49 per 10,000 Aes).
Chun et al., 2021 [24].	Descriptive Epidemiological	Female judo had the highest SRC incidence (1.92 per 1000 Aes). Females had higher SRC rates in all comparable sports (judo, soccer, basketball and volleyball), apart from wrestling.
Bretzin et al., 2018 [25].	Descriptive Epidemiological	Females were 1.9× more likely to sustain SRCs in softball/baseball, soccer and basketball. Men’s lacrosse was the only sex-comparable sport with a higher SRC incidence. Females had a longer return-to-play duration post-SRC than males (13.8 vs. 12 days, *p* < 0.001)
Covassin et al., 2018 [26].	Retrospective Cohort	Men’s football had the highest SRC incidence followed by men’s ice hockey (4.94% and 3.76%, respectively). In sex-comparable sports, females experienced higher SRC rates, e.g., in soccer (3.04% vs. 1.85%) and basketball (2.92% vs. 1.15%), but not in lacrosse (1.08% vs. 1.72%).
Van Pelt et al., 2021 [27].	Meta-Analysis	Females experienced 3.76 SRC per 10,000 Aes, compared to males experiencing 3.65 (*p* < 0.001). Females experienced more SRCs in soccer (*p* < 0.001), basketball (*p* < 0.001), volleyball (*p* < 0.001) and basketball/softball (*p* < 0.01). Males experienced more SRCs in lacrosse (*p* < 0.01).
Cheng et al., 2019 [28].	Meta-Analysis	Females had a higher rate of SRCs in soccer (*p* < 0.01) and basketball (*p* < 0.001). Females experienced a statistically insignificantly higher rate of SRCs in baseball/softball, swimming/diving and track and field. Males experienced an insignificantly higher rate in ice hockey and lacrosse.

^1^ Abbreviations: AE, athletic exposure.

**Table 2 brainsci-13-01310-t002:** Summary of studies examining sex-based vulnerability for SRC.

Author(s), Year	Study Design	Key Findings
McGroaty et al., 2020 [29].	Systematic Review	Females may have lower biomechanical tolerance for head impacts and are more likely to report symptoms. Menstrual cycle phase and hormonal contraceptive use were not associated with shorter symptom duration; however, less severe symptoms occur with hormonal contraceptive use.
Esponeka et al., 2020 [30].	Retrospective Cross-sectional	Females had smaller overall neck circumferences (*p* < 0.001) and neck circumferences in proportion to BMI (*p* < 0.001). There was no significant association between overall or proportional neck circumference and history of SRC.
Huber et al., 2021 [31].	Prospective Observational	Females had lower head impact rates in soccer (1.41 vs. 3.08 per AE ^1^), basketball (0.25 vs. 0.90 per AE) and lacrosse (0.83 vs. 0.06 per AE).
Ling et al., 2020 [32].	Systematic Review and Meta-Analysis	Female athletes are at higher risk of SRC due to ball/equipment contact in lacrosse (*p* < 0.001), soccer (*p* < 0.001) and while heading the ball in soccer (*p* < 0.001). However, they are less likely to sustain SRCs due to player contact in lacrosse (*p* < 0.001), basketball (*p* = 0.01), ice hockey (*p* < 0.001), soccer (*p* < 0.001) and heading in soccer (*p* < 0.001).
Gallagher et al., 2018 [33].	Retrospective Cohort	There was no significant sex difference in symptom severity, but females had a significantly longer TTR ^1^ (22 days vs. 13 days, *p* < 0.05). Females using hormonal contraceptives had lower symptom severity (*p* < 0.05), but there was no effect on TTR.
Beidler et al., 2021 [34].	Cross-sectional	Female sex was associated with an odds ratio of 1.77 of experiencing high levels of anxiety regarding SRC.
Schmitt et al., 2021 [35].	Cross-sectional	Females reported significantly more traits (anxiety, clarity, symptom variability and perception of control) associated with higher anxiety regarding SRC. Females also had a greater understanding of SRC.
McAllister-Deitrick et al., 2021 [36].	Cross-sectional	Females had a significantly higher level of knowledge regarding SRC (*p* < 0.001) and symptoms of SRC (*p* < 0.01). Female athletes were significantly less likely to not disclose SRC (*p* < 0.01) or continue to play with SRC (*p* < 0.01).
Callahan et al., 2022 [37].	Cross-sectional	Females reported lower brief sensation-seeking scores (*p* < 0.01), which were associated with a higher intention to disclose SRC symptoms (*p* < 0.01) and a higher likelihood of a history of SRC disclosure (*p* < 0.01).
Milroy et al., 2021 [38].	Cross-sectional	Latent profile analysis used to determine athletes’ profiles in regard to conclusion disclosure revealed that females are less likely to be in a high-risk group for concussion non-disclosure (*p* < 0.001).
Sullivan and Molcho, 2021 [39].	Survey	Females were less likely to express concern regarding the negative consequences of SRC disclosure, e.g., harming team performance (*p* < 0.001) and teammates thinking less of the player (*p* < 0.001). Females were more likely to report symptoms such as vomiting (*p* < 0.001), memory problems (*p* = 0.001) and light/noise sensitivity (*p* = 0.011).
Anderson et al., 2020 [40].	Cross-sectional	Females expressed a significantly higher intention to seek care for concussion symptoms (*p* < 0.04), which was weakly correlated with intention to seek care for SRC.
Beran et al., 2022 [41].	Systematic Review	The literature suggests that females have a higher level of knowledge regarding SRC and are more likely to report concussion symptoms.
Anderson et al., 2021 [42].	Cross-sectional	In a multivariant analysis, male sex was one of the factors associated with SRC non-disclosure; however, female sex was not implicated as a significant factor for non-disclosure.
Kay et al., 2021 [43].	Mixed-Methods Parallel Research	No significant sex difference was identified in concussion knowledge, attitudes, beliefs or reporting intentions.
O’Connor et al., 2021 [44]	Cross-sectional	No significant sex differences were identified in total SRC knowledge, recognition of signs and symptoms, perceived pressure to return to play prior to recovery or concussion non-disclosure.
Weber et al., 2019 [45]	Cross-sectional	Females had a higher indirect SRC reporting intention (*p* = 0.035); however, there was no sex difference in direct-reporting intentions or behaviors.
Zynda et al., 2021 [46]	Descriptive Epidemiological	Females were 1.26× less likely to be removed from play following SRC in all sports studied—soccer, lacrosse, swimming/diving, baseball/softball and track and field.

^1^ Abbreviations: AE, athletic exposure; TTR, time to recover.

**Table 3 brainsci-13-01310-t003:** Summary of studies examining sex differences in concussion assessment and neurocognitive testing.

Author(s), Year	Study Design	Key Findings
Petit et al., 2020 [47].	Cross-sectional	At baseline, there was no sex differences observed in any component of SCAT5 ^1^.
Norheim et al., 2018 [48].	Retrospective Cross-sectional	At baseline, females performed better on immediate and delayed recall when testing using the 10-word list component of SCAT5. However, average scores did not vary greatly between males and females.
Hutchison et al., 2021 [49].	Cross-sectional	Females performed better than males on immediate and delayed recall when tested using the 10-word list component of SCAT at baseline, and a higher percentage of females achieved a perfect score on delayed recall (11.1% vs. 4.3%).
Bailey et al., 2022 [50].	Prospective cohort	When SCAT5 was administrated twice at baseline, females achieved higher symptom scores at both administrations (*p* < 0.05) and performed better on SCAT5 SAC ^1^ (*p* < 0.001). Use of sex normative data did not improve accuracy when using SAC RCI (*p* < 0.01) but did improve when using a low threshold score (*p* < 0.01). However, use of SAC RCI ^1^ (either with or without sex-normative data) was more accurate at detecting SRC than a low SAC score.
Covassin et al., 2020 [51].	Case-control	Concussed female athletes reported higher symptom scores at day 0 post-concussion (*p* < 0.001) when testing using SCAT3; there was no significant sex difference at other time points. There was no sex difference observed on the total SAC score between males and females at any time point.
Hurtubise et al., 2018 [52].	Retrospective chart review	At baseline, females performed better on cognitive scores, including attaining higher SAC (*p* < 0.05), delayed recall (*p* < 0.001) and orientation (*p* < 0.05) scores when assessed using SCAT3. Only 2.3% of females reported neck pain at baseline, compared to 15.6% of males (*p* < 0.05). Post-SRC females performed better on average on orientation (*p* < 0.05) and reported lower severity of blurred vision (*p* < 0.05) and irritability (*p* < 0.05).
Merritt et al., 2019 [53]	Retrospective cohort	No significant sex difference was observed in NCT when individual components of testing batteries were compared (including ImPACT ^1^). Females were significantly more cognitively impaired following SRC (*p* = 0.045) when a summary of all testing batteries was used.
Tsushima et al., 2021 [54].	Cross-sectional	At baseline, females performed significantly better on the visual motor speed component of ImPACT. There was no significant difference observed at baseline or post-SRC or differences between scores at baseline compared to post-SRC.
Covasin et al., 2012 [55].	Prospective cohort	Females performed worse on visual memory on ImPACT (mean 65.1% vs. 70.1%, *p* = 0.049) and reported more symptoms (14.4 vs. 10.10, *p* = 0.035).
Le et al., 2021 [56].	Cross-sectional	No significant sex differences were observed with King–Devick testing at baseline or post-SRC.
Sicard et al., 2018 [57]	Cross-sectional	Females had worse reaction times on Cogstate battery tests examining attention (*p* < 0.01) and executive function (*p* < 0.01). Females with a past SRC performed worse on tests of executive function with 2-back conditions to increase cognitive load (*p* < 0.001), which is not included in the standard Cogstate assessment.
Oldham et al., 2020 [58]	Prospective longitudinal	No sex differences were observed in the assessment of tandem gait at baseline or acutely following SRC.
Howell et al., 2020 [59].	Prospective longitudinal	No sex differences were observed in single-task gait recovery post-SRC; however, females had a slower dual-task gait recovery (*p* = 0.02).
Lumba-brown et al., 2020 [60].	Retrospective review	Females affected by SRC performed worse on VOMS ^1^ measures including smooth pursuit (*p* = 0.045), convergence (*p* = 0.031) and visual motor sensitivity (*p* = 0.045).
Studenka and Raikes et al., 2020 [61].	Quasi-experimental	Females with a history of previous SRC performed better on visual motor tracking tasks (*p* = 0.005) than males. However, females who had previously had more than two SRCs performed worse than males who had a history of more than two SRCs (*p* = 0.031).
Morissette et al., 2020 [62].	Quasi-experimental	No sex differences were observed in cardiopulmonary response at the rest or early stages of the Buffalo concussion treadmill test.
Balestrini et al., 2021 [63].	Longitudinal cohort	Concussed females showed a reduction in heart rate variability compared to non-concussed females while seated (*p* = 0.04). No such reduction was observed in males.

^1^ Abbreviations: SCAT5, Sport Concussion Assessment Tool 5; SAC, Standardized Assessment of Concussion; RCI, Reliable Chance Index; NCT, neurocognitive testing; ImPACT, Immediate Post-Concussion and Cognitive Testing; VOMS, Vestibular Oculomotor Motor Screening.

**Table 4 brainsci-13-01310-t004:** Summary of studies examining sex differences in SRC symptoms and recovery time.

Author(s), Year	Study Design	Key Findings
Hammer et al., 2021 [64].	Prospective cohort	Females reported higher symptom severity 24–72 h after SRC (*p* = 0.004). No sex difference was observed for depressive symptoms.
Stephenson et al., 2023 [65].	Cross-sectional	Females with SRC reported a higher number of symptoms (*p* = 0.001) and more severe (*p* < 0.001) symptoms. Females had higher cognitive/fatigue (*p* = 0.001), anxiety/mood (*p* = 0.001), and ocular (*p* < 0.01) symptom scores.
Chandran et al., 2020 [66].	Descriptive epidemiological	In high-contact sports, more females with SRC reported irritability (15.4% vs. 9.7%, *p* <0.001), but females were less likely to report loss of consciousness (1.2% vs. 4.0%, *p* < 0.001). Fewer females than males playing high-contact sports had symptom resolution after fewer than 7 days (47.7% vs. 58.8%, *p* < 0.001), and a greater proportion took greater than 21 days for symptoms to resolve (7.3% vs. 3.7%, *p* < 0.001) or return to play (9.8% vs. 6.0%, *p* < 0.001). In low-contact sports, a higher percentage of males reported amnesia post-SRC (17.5% vs. 5.8%, *p* < 0.001).
Bretzin et al., 2021 [67].	Descriptive epidemiological	The median time to symptom resolution was 9 days for females and 8 days for males (*p* < 0.001), and the median time to return to academics was 9 days and 7 days, respectively (*p* < 0.001). No significant differences were observed in full or partial return to play.
Wang et al., 2022 [68].	Case–cohort	Females did not take significantly longer to return to play.
Putukian et al., 2021 [69].	Prospective cohort	Female athletes did not have a greater risk of SRC nor a longer recovery time.
Kerr et al., 2018 [70]	Ambispective cohort	No significant sex differences were observed in incidence of post-concussive syndrome.
Tamura et al., 2020 [71].	Cross-sectional	Concussed females took longer to complete a six-step graded exercise program (21.6 vs. 19.3 days), likely due to taking longer (14.7 vs. 13.0 days) to reach step three, which required clearance by a physician.
Willer et al., 2019 [72].	Quasi-experiment	Athletes prescribed rest post-SRC had delayed recovery (15 vs. 13 days, *p* = 0.020) compared to those prescribed an anerobic exercise program. No sex differences in recovery time were observed in concussed athletes prescribed rest, aerobic exercise or a placebo-like stretching program. Females with SRC prescribed rest suffered increased symptoms compared to other groups (*p* = 0.04).
Iverson et al., 2017 [73].	Systematic review	The literature on recovery time and persistent symptoms is mixed; however, it seems to support that overall, concussed females have longer recovery times and may experience symptoms for longer than one month.
Pinto et al., 2017 [74].	Retrospective observational	Female athletes were 3.8 times more likely to be prescribed medication post-SRC.

## Data Availability

No new data were created or analyzed in this study. Data sharing is not applicable to this article.

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
