# Peer review of "Are Sex Differences in Collegiate and High School Sports-Related Concussion Reflected in the Guidelines? A Scoping Review"

_brainsci, 2023, doi:10.3390/brainsci13091310_

Round 1

Reviewer 1 Report

Thank you for submitting your manuscript. The occurrence of sport-related concussion often leads to serious sequelae, and prevention is critical. However, the rationale for conducting this study is insufficient as to why gender differences in SRC were examined in HS and collegiate athletes. In addition, the results are only enumerated and the author's thoughts and the need for future research are unclear. After reviewing the manuscript, I have several questions and wishes, I hope they will be useful in your further work in the article.

Title

If it is in the interrogative form? is not necessary?

Introduction

General:

You should specify why the subjects in this study are Collegiate and High School athletes.

General:

There are too many paragraphs. You should summarize relevant content together and emphasize what you want to convey in each paragraph.

General:

It is difficult to understand the rationale for why it is necessary to investigate sex differences in the SRC.

Page1, line 34

Delete the period before [1].

Page1, line 38

The font type or density may be different. Please check the following text as well.

Methods

General:

Although the focus is on gender differences, it is necessary to take into account the differences between men and women in the competitive population to begin with.

Page 2, line 74-75

‘Sex’ AND ‘Concussion’; ‘Sex’ AND ‘Sport’ AND ‘Concussion’; ‘Gender’ AND ‘Sport’ AND ‘Concussion’; and ‘Gender’ AND ‘Concussion, is there a reason you wrote them in this order?

Results

General:

No core figure or table was found. A review article needs a figure or table that communicates the results in the simplest and clearest way. 

General:

Please make the table presentable. Some have lines, some do not. I feel it would be easier to see the order of the references if they were sorted by Year or Study design. Also, it is recommended that tables not span pages.

General:

Text in Figure1 is not clear and small, you should match the size of other text.

General:

Perhaps, I felt it necessary to describe the figure and result of the characteristics between high school and collegiate as well as the sex differences in SRC.

General:

When indicating table, etc., in the sentences, it is better to use bold type.

Discussion

General:

The discussion has a lot of overlap with the results, and it is unclear what you want to convey from these results. Since it only introduces what is described in the references, it should be clear what can be said from the results, what kind of research is needed in the future, and what is expected by conducting such research.

General:

There are some discrepancies between the introduction and the discussion.

Page 8, line 293

As a principle, the abbreviation NCT should be used throughout the paper after the first mention of neurocognitive testing.

Page 9, line 309

Based on the present results, can you assert that TTR is longer in female? It seems inconsistent with the text of page7, line 249, which you stated that "this is not consistent among all studies."

Page 9, line 337

This research, you did not find a significant difference by sex in the SCR, What is the significance of continuing this study?

General:

The following is a discussion of the summary in the results. What is your main outcome?

Conclusions

General:

The need for future research found in the results of this study should also be briefly described.

Thank you for submitting your manuscript. The occurrence of sport-related concussion often leads to serious sequelae, and prevention is critical. However, the rationale for conducting this study is insufficient as to why gender differences in SRC were examined in HS and collegiate athletes. In addition, the results are only enumerated and the author's thoughts and the need for future research are unclear. Therefore, I cannot recommend publication of this manuscript. After reviewing the manuscript, I have several questions and wishes, I hope they will be useful in your further work in the article.

Author Response

Reviewer #1

Comment 1 (Title): If it is in the interrogative form? is not necessary?

Author’s Response: Thank you for your observation. As the title is phrased as a question, we have now amended it to include a question mark.

Changes to Text: Title revised to “Are Sex Differences in Collegiate and High School Sports-Related Concussion Reflected in the Guidelines? A Scoping Review”

Comment 2 (Introduction): You should specify why the subjects in this study are Collegiate and High School athletes:

Author’s Response: Thank you for comment. We chose to specifically examine sex differences in high school and collegiate athletes as this demographic is most affected by SRC.

Changes to Text: “We opted to focus on investigating sex differences in high school (HS) and collegiate athletes as this is the demographic that is most affected by SRC” (Introduction, page 2, lines 66-67)

Comment 3 (Introduction): There are too many paragraphs. You should summarize relevant content together and emphasize what you want to convey in each paragraph.       

Author’s Response: As suggested, we have reduced the number of paragraphs in the introduction and summarized similar content within the same paragraph.

Changes to Text: Please see changes made to introduction.

Comment 4: (Introduction): It is difficult to understand the rationale for why it is necessary to investigate sex differences in the SRC.          

Author’s Response: Thank you for this observation. We chose to look at sex differences in SRC, as women sports have been gaining increasing popularity and increased rates of participation. However female athletes are broadly underrepresented in the current literature on SRC, therefore we felt it would be warranted to do a comprehensive review on the current evidence on SRC and how this could be addressed by guidelines on managing SRC.

Changes to Text: “During this time female participation in both amateur and professional sports has also been increasing, with females now making up 43.7% of athletes within all divisions of the National Collegiate Athletic Association (NCAA). Alongside the increase in female participation in sports, there is a growing awareness that anatomical, physiological, and biomechanical factors place female athletes of sustaining sports injuries including SRC. Despite this, female athletes are underrepresented in SRC literature, including in consensus statements on SRC” (Introduction, pages 1-2, lines 40-45)

Comment 5: (Introduction, Page 1, Line 34): Delete the period before [1]

Author’s Response: Thank you for this spotting this typo. This has now been amended accordingly.

Comment 6: (Introduction, Page 1, Line 34): The font type or density may be different. Please check the following text as well

Author’s Response: Thank you for noticing this. This line has been removed in our revised manuscript.

Comment 7: (Methods): Although the focus is on gender differences, it is necessary to take into account the differences between men and women in the competitive population to begin with    

Author’s Response: Thank you for your feedback. As mentioned in response to comment 3, we felt it were necessary to investigate the differences in between men and women in sports due to the anatomical, physiological and biomechanical differences between males and females which translate over to differences in sport concussion.

Comment 8: (Methods): ‘Sex’ AND ‘Concussion’; ‘Sex’ AND ‘Sport’ AND ‘Concussion’; ‘Gender’ AND ‘Sport’ AND ‘Concussion’; and ‘Gender’ AND ‘Concussion, is there a reason you wrote them in this order?       

Author’s Response: We listed the search terms used broadly to match the order in which the literature search were carried out. We did not feel it would be necessary to specify this within the final manuscript.

Comment 9: (Results): No core figure or table was found. A review article needs a figure or table that communicates the results in the simplest and clearest way.

Author’s Response: Thank you for your feedback on our results and identifying the need for a core table/figure. We have responded to this comment by expanding on the previous tables to include the key findings of each study. In order to allow the results to be simpler to access for the readers.

Changes to Text: Please see changes made to tables 1,2,3 and 4.

Comment 10: (Results): Please make the table presentable. Some have lines, some do not. I feel it would be easier to see the order of the references if they were sorted by Year or Study design. Also, it is recommended that tables not span pages.        

Author’s Response: Thank you for your observations regarding the borders in our tables. We have resolved this and ensured that the borders now match across all tables. Care has also been taken to ensure that tables are contained within the same page. We acknowledge the comment regarding listing our differences by year or study design, however, believe that our approach of listing references in order they are mentioned within the qualitive results section improves readability for readers and makes it simpler for readers to refer back to the tables for further information following reading the text.

Changes to Text: Please see changes make to layout of the tables found in the text.

Comment 11: (Results): Text in Figure1 is not clear and small, you should match the size of other text.

Author’s Response: Thank you for your astute observation. We have now increased the size of figure 1 and matched the size and font of the remainder of the text.

Changes to Text: Please see figure 1 (Results, page 3)

Comment 12: (Results): Perhaps, I felt it is necessary to describe the figure and results of the characteristics between high school and collegiate athletes as well as the sex differences in SRC.

Author’s Response: We thank you for feedback. We acknowledge that there will be difference in age and physical characteristics between high school and collegiate athletes which may compound any sex differences observed. In addition, collegiate level sports are played at a semi-professional level which may have an effect on the frequency and intensity of practices and games. However, as our work is a scoping review, we did not feel that we would be able to comprehensively cover all the potential factors for sex differences in SRC, and this is stated as one of our limitations (discussion, page 12, lines 384-386)

Comment 13 (Results):  When indicating tables, etc., in the sentences it is better to use bold type.

Author’s Response: Thank you for your comment. We have now ensured to use bold type when referring to the tables and figures in the text.

Changes to Text:  Figure 1 changed to Figure 1 (methods, page 3, line 106. Table 1 changed to Table 1 (results, page 4, line 120). Table 2 changed to Table 2 (results, page 5, line 149). Table 3 changed to Table 3 (results, page 8, line 192). Table 4 changed to Table 4 (results, page 8, line 233)

Comment 14 (Discussion): The discussion has a lot of overlap with the results, and it is unclear what you want to convey from these results. Since it only introduces what is described in the references, it should be clear what can be said from the results, what kind of research is needed in the future and what is expected by conducting said research.

Author’s Response: Thank you for your feedback. We acknowledge that some of the results are briefly re-iterated within the discussion however this is only done in order to allow us to elaborate further on our findings, how these relate to wider literature on the topic, and the implications. We also acknowledge the comments regarding more clarity being required about the type of research needed and how this would further understanding of SRC, and elaborated on this with section “4:3 Limitation and Future Work” (discussion, page 12-13, lines 402-435”

Changes to Text: “We would also implore future studies to look into investigating in greater detail the role of sex differences in cervical muscles on SRC and whether strategies such as neck training programs or use of neuromuscular warm up programs are useful in preventing SRC. Finally, we would also recommend more research on the sex-differences in exercise programs following SRC in order to allow clinicians to understand how best to tailor SRC recovery programs for concussed females.” (Discussion, page 13, lines 429-435).

Comment 15 (Discussion): There are some discrepancies between the introduction and the discussion.

Author’s Response: Thank you for your feedback. While not specified by the reviewer we assume  the discrepancies relate to ‘recognize’, ‘rehabilitate’ ‘return-to-learn/return-to-play’ and ‘retire’ being mentioned in the introduction however not elaborated on with our discussion. Therefore, we have now included brief statements regarding these in the discussion hopefully resolving any discrepancies.

Changes to Text: Please see discussion of ‘recognize’ (discussion, page 12, lines 319-321), ‘rehabilitate’ (discussion, page 12, lines 376-380), ‘return-to-learn’/’return-to-play’ (discussion, page 12, lines 389-391) and retire (discussion, page 12, lines 394-397)

Comment 12 (Discussion, page 8, line 293): As a principle the abbreviation NCT should be used throughout the paper after the first mention of neurocognitive testing.

Author’s Response: Thank you for spotting this inconsistency. This has now been changed.

Changes to Text:  “Neurocognitive testing batteries” changed to “NCT batteries” (discussion, page 11, line 361)

Comment 13 (Discussion, page 9, line 309): Based on the present results, can you assert that TTR is longer in female? It seems inconsistent with the text of page7, line 249, which you stated that "this is not consistent among all studies.”

Author’s Response: Thank you for your comment. We agree that these two statements conflict with each other, therefore have re-phrased this line to clarify that while females do have a longer TTR in most studies, this was not a consistent finding among all studies analysed.

Changes to Text:  “As previously mentioned in this study we found that concussed females appear to have a longer TTR” changed to “As previously mentioned, while not consistent across all studies, females seem to have a longer TTR” (discussion, page 12, 379-380.

Comment 14 (Discussion, page 9, line 337): This research, you did not find a significant difference by sex in the SCR, What is the significance of continuing this study?

Author’s Response: Thank you for comment. We would disagree with the reviewers claim that we found no significant sex differences in SRC. In our study we have shown that women experience higher rates of SRC compared to males in sex comparable sports. We have also shown that there is evidence of biomechanical factors (neck strength and size), hormonal influences and social factors placing females at higher risk of SRC/more likely to disclose SRC. We have also shown that most studies agree that females experience different and more severe symptoms and take longer to recover. Therefore, we feel further research is needed to fully examine why females are at greater risk sustaining SRC and suffering from more severe and longer-lasting symptoms in order to help mitigate this risk and provide evidence-based guidelines for management of concussed females. Finally, we also encourage future studies on SRC to aim to recruit equal recruitment of males and females to prevent female athletes being underrepresented in concussion research.

Comment 15 (Discussion): The following is a discussion of the summary in the results. What is your main outcome?

Author’s Response: Thank you for the comment. In response to other reviewers’ suggestions, we have included our primary and secondary outcomes within the introduction (introduction, page 2, line 59-64). Our primary outcome was to examine sex difference in SRC and how these relate to epidemiology, sex-based risk factors for SRC, sex differences in concussion assessment, and sex differences in symptoms and TTR and our secondary outcome was to examine how any sex differences we examined may related to the CISG most recent consensus statement. These outcomes are discussed within section “4:1: Sex differences in Sport-Related Concussion” (discussion, page 10, lines 262-309) and section “4:2: Are sex differences reflected in the Amsterdam consensus” (discussion, pages 11-12, lines 314-400).

Comment 16 (Conclusion): The need for future research found in the results of this study should also be briefly described.

Author’s Response: Thank you for this suggestion. We have included a brief mention of need for future research within the conclusion.

Changes to Text:  “Future research is encouraged in order to unravel the complex biomechanical, hormonal and social influences which are implicated in sex differences in SRC. Greater understanding of these will hopefully allow for evidence-based prevention and management protocols for female athletes. We also encourage future work to focus on im-proving female representation within SRC research.” (Conclusion, page 13, lines 446-451).

Reviewer 2 Report

Thank you for this contribution to the literature. An interesting study with clear implications highlighted. My only comment is on the introduction. Could more detail be provided to the research area and SRC in general to provide context for readers. At current, the introduction is very shallow and only focuses on the CISG consensus. Also expand on the sex/gender terminology differences. Gender is more than an 'individuals identity'. It is a social concept that refers to a range of social processes and behaviours which relate to sex. This is important as there are a lot of people who would read your paper and suggest any differences in SRC outcomes between men and women are more social than biological. 

Author Response

Reviewer #2:

Comment 1: Thank you for this contribution to the literature. An interesting study with clear implications highlighted. My only comment is on the introduction. Could more detail be provided to the research area and SRC in general to provide context for readers. At current, the introduction is very shallow and only focuses on the CISG consensus. Also expand on the sex/gender terminology differences. Gender is more than an 'individuals identity'. It is a social concept that refers to a range of social processes and behaviours which relate to sex. This is important as there are a lot of people who would read your paper and suggest any differences in SRC outcomes between men and women are more social than biological.

Author’s Response: Thank you for providing insightful comments regarding our introduction. In line with reviewers’ suggestions, we have re-written the introduction to provide more detail on the research area, we have also elaborated in greater detail on the difference between sex and gender and how these distinctions specifically relate to concussion research.

Changes to Text:  Please see changes made to the introduction. More detailed discussion on the SRC can be found in pages 1-2, lines 33-45. More detailed discussion on sex and gender can be found in page 2, lines 68-73.

Reviewer 3 Report

Thank you for the opportunity to review this study, my considerations are below:

Title:

- Is the title in question format? If yes, included the question mark (?) placed in place of the colon (:)

Abstract:

- For a better understanding of the readers, I suggest that the authors include the subtopics in the abstract (introduction, objectives, methods, results and conclusion)

- The purpose of the study is partially implied, however, I suggest to the authors directly mentioned, what was the purpose of the study.

- It is not clear whether the authors intend to look for injury rates or other concussion-related outcomes between sexes. I believe that the authors could be more direct in what they intended to seek, to improve readers' understanding of the text of this article.

- I suggest to the authors mentioned more about the methods. Was there a language restriction on searches? Was only Medline used in the searches? Pubmed is the access platform, the database is Medline.

- The conclusion of the study cannot be that more studies need to be carried out on the topic. The conclusion of the study must answer the purpose of the study.

- I suggest that the authors rewrite the abstract, perhaps the inclusion of subtopics will help the authors to make the text more understandable.

Introduction:

- The introduction is very confusing and does not present an outline of the study problem with a logical sequence, I suggest that the authors better rewrite the introduction, better organizing the paragraphs so that the text presents the research problem in a logical chronological order.

- The authors use the terms “sex” and “gender” which are not the same thing. Sex is what you are born with, and gender is what you understand yourself as a person, so I can have a certain sex and not identify/recognize with it, that is gender. I suggest that authors be careful when using the term gender.

- The purpose is not clear, when the authors mention the term “sex differences in concussion” this is too broad. Do they want to analyze the occurrence of injuries? What are the symptoms? How long do the symptoms last? Specify the occurrence of injuries due to sports practiced? So, the way it is written, the objective of the room for different interpretations, which is not correct, the objective of the study needs to be clear. The authors need to improve the description of the purpose of the study, if applicable, mention primary and secondary objectives to improve the description of what the authors intended to research in this study.

Methods:

- I believe that one of the questions is poorly reformulated. What are the sex differences regarding sports-related concussions? Also, like the objective, this question is poorly worded. Authors need to better specify and describe what they are looking for.

- What types of studies would be eligible for this review? All?

- Why were other important databases not searched? This is a serious limitation of this review. After all, there may be other studies published in other important databases such as (EMBASE, SCOPUS, Web of Science, PEDro, CINAHL, LILACS), which could perhaps change the results of this conclusion. I suggest adding this as a limitation of this study.

- I suggest also adding the language restriction as another limitation of this review.

- Why was there a publication time restriction? The authors do not mention either in the introduction or in the methods because it would be important to restrict the publication time of the studies, this is another limitation of this study.

Results:

- These tables don't make sense to me. They don't bring any relevant information.

- I suggest that the results expressed in tables are the percentages that the authors bring in the text of the results.

- These are the actual results of this work and the most important ones, so they have to be easily accessible and more visible to readers.

- Thus, I suggest that authors delete the current tables, or leave at least the first one and include new tables by theme, for example, table 2: occurrence of injuries related to concussions and the tables contain information with their respective percentages and authors. I believe that the results will be more exposed and more visible to readers and the results of the current tables may be in the texts, as they are less important.

Discussion:

- The discussion was a lot about the 11 “Rs” however, I missed how to prevent these disturbances.

- What can the team/club health team do to prevent or minimize these disturbances?

- I missed a more robust discussion on top of the main findings of the study. For example, there is a discussion that women's cervical muscles are more phasic, perhaps that's why they have more post-concussion symptoms. How could this be improved in terms of performance or prevention? I missed that kind of approach in the discussion.

- I also missed a greater discussion on vestibular rehabilitation, its inclusion in teams/clubs and a paragraph informing about its importance, because the concussion causes a lot of otoneurological and vegetative symptoms, such as dizziness, vertigo, vomiting, vestibular migraine, mentioned in the introduction of the study. Thus, these aspects could be further explored in the discussion.

- I suggest including the other limitations discussed above in the article.

Conclusion:

- This conclusion is much better than the conclusion of the abstract. Rearrange the conclusion of the abstract by this one.

Author Response

Reviewer #3:

Comment 1 (title): Is the title in question format? If yes, included the question mark (?) placed in place of the colon (:)

Author’s Response: Thank you for this observation. We have amended the title to include a question mark in place of the colon.

Changes to Text:  Title of paper changed to “Are Sex Differences in Collegiate and High School Sport-Related Concussion Reflected in the Guidelines? A Scoping Review

Comment 2 (abstract): For a better understanding of the readers, I suggest that the authors include the subtopics in the abstract (introduction, objectives, methods, results and conclusion)

Author’s Response: Thank you for your feedback. We have now included the following subsections in our abstract: background, objectives, methods, results and conclusion.

Changes to Text:  Please see changes to abstract.

Comment 3 (abstract): The purpose of the study is partially implied, however, I suggest to the authors directly mentioned, what was the purpose of the study.

Author’s Response: Thank you for your comment. Our abstract now contains a subsection titled ‘objectives’ which implicitly states our objectives.

Changes to Text:  “Objectives: We aimed to identify sex differences in epidemiology, clinical manifestation, and assessment of SRC and examine how these relate to the 6th international consensus on concussion in sport (ICCS).” (Abstract, page 1, lines 14-15).

Comment 4 (abstract): It is not clear whether the authors intend to look for injury rates or other concussion-related outcomes between sexes. I believe that the authors could be more direct in what they intended to seek, to improve readers' understanding of the text of this article.

Author’s Response: Thank you for your observation. We have now included a brief statement within the objectives as discussed above in which we directly state that we were only looking at sex differences in SRC.

Comment 5 (abstract): I suggest to the authors mentioned more about the methods. Was there a language restriction on searches? Was only Medline used in the searches? Pubmed is the access platform, the database is Medline.

Author’s Response: Thank you for these suggestions. We have amended the methods contained within the abstract according to this feedback to clarify that there was a language restrictions, and have corrected the terminology in respect to PubMed/MEDLINE

Changes to Text:  “Methods: We conducted a scoping review of the Medline database and identified 58 studies examining the effects of sex on SRC in collegiate and high school athletes that were written in English and published in a peer-reviewed journal between March 2012 – March 2022.” (Abstract, page 1, lines 16-19)

Comment 6 (abstract): The conclusion of the study cannot be that more studies need to be carried out on the topic. The conclusion of the study must answer the purpose of the study.

Author’s Response: Thank you for this observation as well as comment #25. We have taken this into consideration and re-written the conclusion contained in the abstract.

Changes to Text:  Conclusion: Females are at greater risk and experience SRC differently compared to males; this is mostly likely due to a combination of biomechanical factors, differences in neck musculature, and hormonal and social factors. Sex differences are not widely addressed by the 6th ICSS which in-forms many sports’ concussion protocols.(Abstract, page 1, lines 23-26)

Comment 7 (abstract): I suggest that the authors rewrite the abstract, perhaps the inclusion of subtopics will help the authors to make the text more understandable.

Author’s Response: Thank you for this your suggestion. We have re-written the abstract with respect to comments 1-6.

Changes to Text:  Please see changes to abstract. (Abstract, page 1, lines 12-26).

Comment 8 (introduction): The introduction is very confusing and does not present an outline of the study problem with a logical sequence, I suggest that the authors better rewrite the introduction, better organizing the paragraphs so that the text presents the research problem in a logical chronological order.

Author’s Response: Thank you for this your comment. We have re-structured our introduction with this comment in mind and hope you find the revised introduction to be presented in a more organized and logical way.

Changes to Text:  Please see changes to introduction (Introduction, pages 1-2, lines 33-73)

Comment 9 (introduction): The authors use the terms “sex” and “gender” which are not the same thing. Sex is what you are born with, and gender is what you understand yourself as a person, so I can have a certain sex and not identify/recognize with it, that is gender. I suggest that authors be careful when using the term gender.

Author’s Response: Thank you for this your feedback. We acknowledge that ‘sex’ and ‘gender’ are terms that refer to different aspects of a persons identify, and the importance of using the correct terminology. We have expanded on this distinction in our introduction, to make clear that we are using the term sex as we are focusing on biological differences between males and females. However, we also acknowledge that gender is important in the context of SRC.

Changes to Text:  “Within the literature on SRC, gender and sex are often used interchangeably, however, it is important to distinguish that the term ‘sex’ is used to refer to the biological differences between males and females, whereas ‘gender’ refers to socially con-structed roles, behaviour and identities [12]. There is a recognition that sex and gender both play a role in concussion [13], however, in this review, we were primarily interested in investigating biological differences in SRC therefore the term sex is used throughout.” (Introduction, page 2, lines 68-73)

Comment 10 (introduction): The purpose is not clear, when the authors mention the term “sex differences in concussion” this is too broad. Do they want to analyze the occurrence of injuries? What are the symptoms? How long do the symptoms last? Specify the occurrence of injuries due to sports practiced? So, the way it is written, the objective of the room for different interpretations, which is not correct, the objective of the study needs to be clear. The authors need to improve the description of the purpose of the study, if applicable, mention primary and secondary objectives to improve the description of what the authors intended to research in this study.

Author’s Response: Thank you for your feedback. We have revised our introduction to include a clear primary and secondary aim.

Changes to Text:  “Our primary aim was to identify the sex differences in the epidemiology of SRC, underlying reasons why females may be more vulnerable to SRC, differences in symptoms experienced and time to recover from concussion between male and female athletes and finally sex differences in concussion assessment. Our secondary aim was to examine whether any sex differences we identified were addressed by the CISG’s 6th consensus statement.” (Introduction, page 2, lines 59-64)

Comment 11 (methods): I believe that one of the questions is poorly reformulated. What are the sex differences regarding sports-related concussions? Also, like the objective, this question is poorly worded. Authors need to better specify and describe what they are looking for.

Author’s Response: Thank you for your feedback. We have re-formatted our original research question into four separate questions that explicitly state what we were examining.

Changes to Text: Please see methods, page 2, lines 78-85.

Comment 12 (methods): What types of studies would be eligible for this review? All?

Author’s Response: Thank you for your observation. We have now included a brief sentence stating that case reports and narrative reviews were not included in our study.

Changes to Text: “Exclusion criteria were: (1) Only single sport studied; (2) Concussions acquired outside of sports; (3) Severe traumatic brain injuries, (4) Imaging or biomarker studies, & (5) Case reports and narrative reviews.” (Methods, page 3, lines 99-100).

Comment 12 (methods): Why were other important databases not searched? This is a serious limitation of this review. After all, there may be other studies published in other important databases such as (EMBASE, SCOPUS, Web of Science, PEDro, CINAHL, LILACS), which could perhaps change the results of this conclusion. I suggest adding this as a limitation of this study.

Author’s Response: Thank you for your comment. We appreciate that since we solely used one database to identify studies, this needs to be highlighted as a major flaw in our methodology. We have therefore included a statement in our limitations to reflect this.

Changes to Text: “Another limitation of our study was that only the Medline database was searched for relevant articles, hence there is a likelihood that studies indexed in other databases were omitted, potentially affecting data and subsequent conclusions” (Discussion, page 12, lines 404-408).

Comment 13 (methods): I suggest also adding the language restriction as another limitation of this review.

Author’s Response: Thank you for your observation. We have added language restriction among the other limitations of our study.

Changes to Text: “However, as concussions occur worldwide it may be difficult to comment on sex differences in concussions in countries where there may be differences in sports played, management principles and sociodemographic differences. This limitation was likely further exacerbated due to our methodology only including studies written in English.” (Discussion, page 13, line 421-422).

Comment 14 (methods): Why was there a publication time restriction? The authors do not mention either in the introduction or in the methods because it would be important to restrict the publication time of the studies, this is another limitation of this study.

Author’s Response: Thank you for your comment. We opted to restrict studies included to those published between March 2012 and March 2022, as in our preliminary search of the data we identified that interest in sex differences in SRC has been recent and as there are few studies older than 10 years old on this topic. Therefore, we decided to have a time restriction in order to allow us to focus on the more recent data. March 2022 was chosen as the cut off point for publication as this is when the study was initially conducted. We have added a brief statement within the limitations in line with you suggestion.

Changes to Text: “Finally, we only included studies published between March 2012 and March 2022, therefore we did not examine the full scope of the literature on sex differences in SRC.” (Discussion, page 12, lines 407-408).

Comment 15 (results): These tables don't make sense to me. They don't bring any relevant information. I suggest that the results expressed in tables are the percentages that the authors bring in the text of the results. These are the actual results of this work and the most important ones, so they have to be easily accessible and more visible to readers. Thus, I suggest that authors delete the current tables, or leave at least the first one and include new tables by theme, for example, table 2: occurrence of injuries related to concussions and the tables contain information with their respective percentages and authors. I believe that the results will be more exposed and more visible to readers and the results of the current tables may be in the texts, as they are less important.

Author’s Response: Thank you for your through feedback on our results. We have taken this on board and re-structured our tables to now include key findings from each study. We hope this improves the readability and accessibility of the results for readers.

Changes to Text:  Please see changes made to tables 1,2,3 and 4.

Comment 16 (discussion): The discussion was a lot about the 11 “Rs” however, I missed how to prevent these disturbances. What can the team/club health team do to prevent or minimize these disturbances?

Author’s Response: Thank you for your comment. We acknowledge that prevention of SRC is key. We discussed prevention strategies suggested within the 6th consensus (discussion, page 11, lines 333-338)  however in light of this comment and comment 17 we have included additional discussion of neck strength in regards to modifying SRC risk.

Changes to Text:  “As previously discussed, sex differences in cervical muscles may place females at a biomechanical disadvantage in regards to head impacts, therefore it would be reasonable to suggest that this risk factor could be modified through neck strengthening pro-grams. However, in the consensus further research is encouraged on this topic, and there are no recommendations made for neck strengthening programs for male or females athletes” (discussion, page 11, lines 333-338).

Comment 17 (discussion): I missed a more robust discussion on top of the main findings of the study. For example, there is a discussion that women's cervical muscles are more phasic, perhaps that's why they have more post-concussion symptoms. How could this be improved in terms of performance or prevention? I missed that kind of approach in the discussion.

Author’s Response: Thank you for your suggestion. We have expanded on some more nuanced sex differences in cervical muscles that may have an effect on biomechanical tolerance to head impact. We also acknowledge the comment regarding improvement of performance and prevention of SRC through neck strengthening programs and have added a brief comment regarding the need for further research on how neck strength may be a modifiable factor in SRC, which is discussed in reply to comment 16.

Changes to Text:  “Despite this it is interesting to note that studies focusing on the role of cervical muscles during soccer heading have shown that players with smaller neck size and strength experience greater head acceleration, this may in part explain why women’s soccer was frequently found to have the highest rate of concussion out of all sex-comparable sports. There is however evidence that neck muscle size and strength is not the only factor involved in responding to head impact with some evidence showing that there is a difference between how male and female cervical muscles react to head impacts. Females have been shown to activate their sternocleidomastoid muscles earlier in response to impact resulting in greater peak angular acceleration and displacement” (Discussion, page 10, lines 270-279 & “One potential explanation for prolonged TTR is females reporting a higher number and greater severity of symptoms, due to factors mentioned above such as lower biomechanical tolerance to impacts and hormonal factors, and thus due to this there is a larger deficit to recover from” (Discussion page 10, 308-309).

Comment 18 (discussion): I also missed a greater discussion on vestibular rehabilitation, its inclusion in teams/clubs and a paragraph informing about its importance, because the concussion causes a lot of otoneurological and vegetative symptoms, such as dizziness, vertigo, vomiting, vestibular migraine, mentioned in the introduction of the study. Thus, these aspects could be further explored in the discussion.

Author’s Response: Thank you for your suggestion. While we agree that vestibular rehabilitation is important in manging vestibular symptoms following SRC, and it’s implementation by team’s is important in supporting athletes’ recovery. However, in our study we did not identify any sex difference in vestibular symptoms experienced post-concussion therefore do not feel it would fit our findings to discuss vestibular rehab and have therefore have only added a brief statement regarding this.

Changes to Text:  “Within ‘Rehabilitation’ the CISG recommends cervicovestibular for neck pain and/or headaches persisting for longer than 10 days, and vestibular rehabilitation for symptoms such as dizziness/balance problems. In our study we did not find any consistent sex differences in neck pain, headaches or vestibular symptoms, which would suggest females would benefit more or less from cervicovestibular or vestibular rehabilitation.” (Discussion, page 12, lines 376-380).

Comment 19 (discussion): I suggest including the other limitations discussed above in the article.

Author’s Response: Thank you for your suggestions, this has been addressed above in regard to comments 12-14.

Comment 20 (Conclusion): This conclusion is much better than the conclusion of the abstract. Rearrange the conclusion of the abstract by this one.

Author’s Response: Thank you for kind comments. This has been addressed in comment 6.

Round 2

Reviewer 3 Report

Congratulations to the authors, they did an excellent job. The article is now much more structured and I believe it is clearer for readers to understand.